# Prediction of ferroelectricity-driven Berry curvature enabling charge- and spin-controllable photocurrent in tin telluride monolayers

Jeongwoo Kim [1,2,7], Kyoung-Whan Kim [3,4,7], Dongbin Shin[1], Sang-Hoon Lee[5], Jairo Sinova[4,6], Noejung Park [1] & Hosub Jin[1]

In symmetry-broken crystalline solids, pole structures of Berry curvature (BC) can emerge, and they have been utilized as a versatile tool for controlling transport properties. For example, the monopole component of the BC is induced by the time-reversal symmetry breaking, and the BC dipole arises from a lack of inversion symmetry, leading to the anomalous Hall and nonlinear Hall effects, respectively. Based on first-principles calculations, we show that the ferroelectricity in a tin telluride monolayer produces a unique BC distribution, which offers charge- and spin-controllable photocurrents. Even with the sizable band gap, the ferroelectrically driven BC dipole is comparable to those of small-gap topological materials. By manipulating the photon handedness and the ferroelectric polarization, charge and spin circular photogalvanic currents are generated in a controllable manner. The ferroelectricity in group-IV monochalcogenide monolayers can be a useful tool to control the BC dipole and the nonlinear optoelectronic responses.

[1] Department of Physics, Ulsan National Institute of Science and Technology, Ulsan 44919, Korea. [2] Department of Physics, Incheon National University, Incheon 22012, Korea. [3] Center for Spintronics, Korea Institute of Science and Technology, Seoul 02792, Korea. [4] Institute of Physics, Johannes Gutenberg University Mainz, Mainz 55099, Germany. [5] Korea Institute for Advanced Study, Seoul 02455, Korea. [6] Institute of Physics, Academy of Sciences of the Czech Republic, Cukrovarnická 10, 162 53 Praha 6, Czech Republic. [7] These authors contributed equally: Jeongwoo Kim, Kyoung-Whan Kim. Correspondence and requests for materials should be addressed to H.J. (email: hsjin@unist.ac.kr)

The concept of Berry curvature (BC) is becoming increasingly pertinent due to its central role in various topological phases and unusual transport phenomena[1–5]. Under symmetry-broken environments in crystalline solids, BC can emerge from the quantum geometry embedded in the electronic structure. It provides an effective magnetic field in momentum space and deforms the electron motion in real space, which becomes a primary origin of exotic transport properties, including various Hall effects[6–13]. As a representative example, when time-reversal symmetry is broken, the anomalous Hall effect arises from the net flux of the BC, which is known as the Berry phase[4,12,14,15]. Recently, the dipole component of the BC that can be induced by inversion asymmetry has been attracting increasing attention due to its potential for optoelectronic applications[16–19]. Under out-of-equilibrium electron distributions, the BC dipole can enable nonlinear optoelectronic transport, which has been realized in photogalvanic experiments[20,21].

Since level crossing can generate a singular BC distribution, topological materials that possess small inverted band gaps or band crossings have been mostly studied as efficient platforms for hosting a large BC dipole[20–26]. For instance, a small-gap quantum spin Hall $WTe_2$ monolayer shows a large inter-band BC and its dipole is manipulated by an external electric field, resulting in the circular photogalvanic effect[20]. Tilted Weyl semimetals and pressurized BiTeI that is driven towards the topological phase transition regime also exhibit a large enhancement in the intra-band BC dipole, leading to the nonlinear Hall effect by generating a transverse photocurrent under linearly polarized light[24,25]. Despite the large BC dipoles in these topological materials, the prominent nonlinear optical properties are available only in response to low-frequency fields due to the small size of the band gap and the subband energy splitting. For high-frequency applications, the system requires a larger band gap corresponding to a higher photon energy. However, realizing a large BC dipole is challenging in large-band-gap systems because a large-band gap impedes singular band inversion and crossing. Therefore, it is desirable to identify a new mechanism for producing a large amount of BC and its corresponding dipole, even in the presence of a relatively large-band gap.

For BC engineering in large-band gap systems, we suggest that ferroelectricity can serve as a tool for manipulating the BC distribution by providing an inversion-breaking order parameter. Using first-principles density functional theory (DFT), we demonstrate that the in-plane ferroelectricity in a SnTe monolayer exhibits a large BC distribution with a band gap of ~1 eV, which corresponds to the near-infrared or visible light range. Due to the ferroelectricity, a pair of positive and negative BC peaks is formed, naturally inducing a BC dipole. Microscopically, the ferroelectric displacement develops nearest-neighbor interorbital hopping channels that are otherwise forbidden by the structural symmetry and it efficiently mixes the orbital characteristics of the conduction and valence bands. Consequently, a pair of the opposite orbital angular momentum textures appears and is referred to as the orbital Rashba effect, to which the large BC dipole is primarily ascribed. In addition to the conventional application of the BC dipole for the nonlinear optoelectronics, we present an intriguing approach for controlling the spin and charge photocurrents either separately or simultaneously via ferroelectric polarization in cooperation with photon helicity. Considering the nonvolatile switching of the electric polarization, the large BC and its dipole in large-gapped ferroelectric systems provide a new approach for multifunctional nonlinear optoelectronic and optospintronic applications.

## Results

**Atomic and electronic structure of the SnTe monolayer.** The SnTe monolayer has a $Pmn2_1$ space group due to the in-plane ferroelectricity[27]. It can be considered as a binary version of the phosphorene puckered structure, where Sn and Te atoms undergo opposite displacements along the [100] direction (the x-axis in Fig. 1a)[28–30]. As a result, the SnTe monolayer has an in-plane ferroelectricity of 12.4 μC cm$^{-2}$ (Supplementary Fig. 1). A mirror symmetry ($M_{xz}$) and a glide mirror symmetry (G) exist, as illustrated in Fig. 1a. When the ferroelectric polarization is aligned along the x-axis, the electronic structure of the SnTe monolayer exhibits two valleys near points X and Y[31], which are hereafter referred to as the X and Y valleys, respectively (Fig. 1b). The conduction and valence bands near the Fermi level (−1 to 1 eV) mostly originate from Sn- and Te-5p orbitals, respectively (Supplementary Fig. 2a). The lowest conduction bands (the highest valence bands) are derived from the Sn(Te)-$p_x$ orbital at the X valley, while they are derived from the $p_y$ orbital at the Y valley (Fig. 1b). Along the circular path from the X valley to the Y valley, the atomic orbitals are aligned along the radial direction for the lowest conduction band (~0.7 eV) and along the tangential

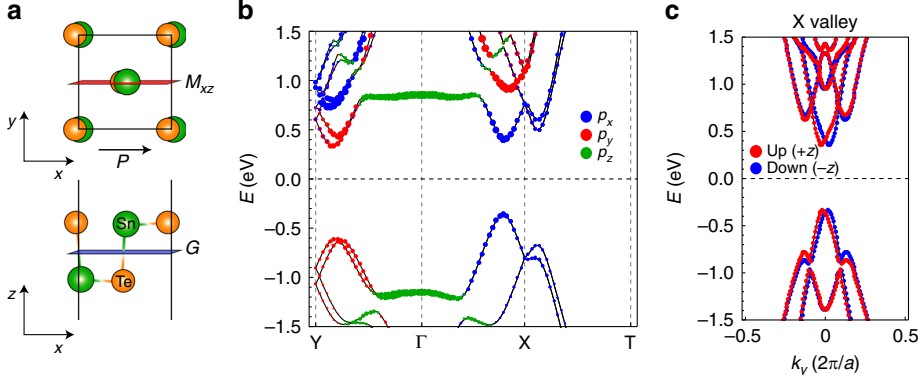

**Fig. 1** Atomic and electronic structure of the SnTe monolayer. **a** Top and side views of the SnTe monolayer. The ferroelectricity is induced by the in-plane atomic movements of the Sn (green spheres) and Te (orange spheres) atoms. $M_{xz}$ and G denote the planes for the mirror symmetry and the gliding symmetry, respectively. **b** The calculated band structure of the SnTe monolayer. The states near the Fermi levels are mainly composed of p orbitals and their weights are plotted in three colors. **c** The calculated band structure for the X valley. The band calculation is performed along the $k_y$-direction for the X valley. Spinup (spindown) states are marked by red (blue) dots. Due to the inversion symmetry breaking induced by the ferroelectricity, a Rashba-type unidirectional spin texture appears in the band structure

direction for the second-lowest band (~1.1 eV) (for details, see Supplementary Fig. 2b–d).

The orbital splitting between the radial and tangential orbitals is described by a simple model Hamiltonian, which, in the following sections, shall play a crucial role in ferroelectrically driven BC dipoles. Conduction electrons near the X valley can be modeled by

$$H_0(\mathbf{k}) = E_X(\mathbf{k}) - J\cos 2\theta_{\mathbf{k}}\tau_z - J\sin 2\theta_{\mathbf{k}}\tau_x, \qquad (1)$$

where $\tau_z = |p_x\rangle\langle p_y| + |p_y\rangle\langle p_x|$, $\tau_z = -i|p_x\rangle\langle p_y| + i|p_y\rangle\langle p_x|$, and $\tau_z = |p_x\rangle\langle p_x| - |p_y\rangle\langle p_y|$ are the pseudospin Pauli matrices in the $p_{x/y}$-orbital basis, $E_X(\mathbf{k})$ is the kinetic energy in the form of $\hbar^2(\mathbf{k} - \mathbf{k}_X)^2/2m$ near the X valley, and $\mathbf{k}_X$ is the position of the X valley. The second and third terms in $H_0(\mathbf{k})$ model momentum-dependent orbital splitting, where $2J > 0$ is the orbital splitting energy, and $\theta_{\mathbf{k}} = \arg(k_x + ik_y)$ makes the orbital structure consistent with the DFT calculations (Supplementary Fig. 2c, d).

Due to the orthorhombic structure induced by the in-plane ferroelectricity, the X and Y valleys are inequivalent to each other. For the X valley, a unidirectional Rashba-type spin splitting is observed along the $k_y$-direction (Fig. 1c), which is generated by the combination of $x$-axis ferroelectric polarization and spin–orbit coupling (SOC)[32–34]. The spin states are degenerate along the Γ–X line due to the mirror symmetry $M_{xz}$. In addition, for the spin expectation value at each spin split-off band, the glide mirror symmetry $G$ allows the out-of-plane component only, as shown in Fig. 1c.

**Ferroelectrically driven BC.** When the ferroelectric polarization breaks the inversion symmetry of the system, a unique polarization-dependent BC distribution can emerge. To elucidate the relation between the ferroelectricity and the BC in the SnTe monolayer, we calculate the BC distribution, namely, $\boldsymbol{\Omega}(\mathbf{k}) = \Omega(\mathbf{k})\mathbf{z}$, over the first Brillouin zone (see Methods). As shown in Fig. 2a, large BC peaks are established at the X valley. Interestingly, a pair of positive and negative BC peaks appears at the X valley, naturally linking to the BC dipole, which is shown later. Moreover, the overall sign changes by the polarization reversal (Supplementary Fig. 3); hence, direct coupling occurs between the BC and the ferroelectricity. Therefore, the ferroelectricity of the SnTe monolayer provides an efficient approach for producing a large BC distribution in a controllable manner.

With varying the magnitude of the ferroelectric polarization ($P$) or the SOC strength ($\lambda$), the BC distribution is investigated in

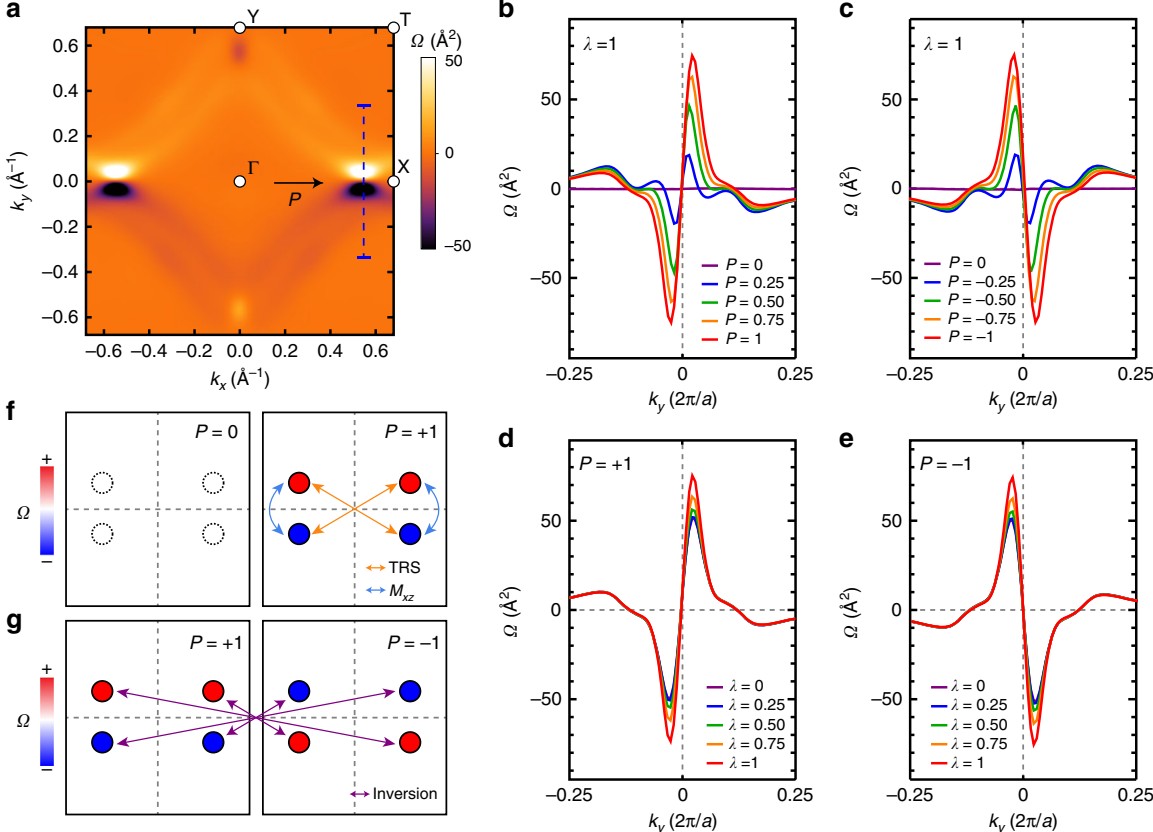

**Fig. 2** Berry curvature ($\Omega$) of the SnTe monolayer. **a** The calculated Berry curvature (BC) map in the first Brillouin zone. The BC dipoles are formed at the X valleys. **b–e** The BC that is calculated along the vertical blue line in (**a**) under varying (**b, c**) ferroelectric polarization magnitude ($P$) and (**d, e**) spin–orbit coupling strength ($\lambda$), where the values of $P$ and $\lambda$ are normalized by the native values. The ferroelectricity reversal from (**b, d**) the positive to (**c, e**) the negative direction changes the sign of the BC distribution. These data imply that the BC dipole is strongly coupled with the ferroelectricity, rather than spin–orbit coupling. **f** Schematic drawings of the BC according to the symmetry of the system. The red and blue circles denote the two opposite signs. The zero BCs are enforced by the combination of the time-reversal symmetry and the inversion symmetry in the absence of the ferroelectricity. When the in-plane ferroelectricity develops, the positive and negative-BC peaks are formed along the $k_y$. The peaks that are connected by orange (blue) arrows denote time-reversal (mirror-symmetric) pairs; they are transformed to each other by time-reversal (mirror) operation. **g** Schematic diagrams of the response of the BC to the ferroelectric switching. Since the $+P$ and $-P$ configurations are inversion partners, the BCs of two configurations are opposite at the same $k$ point. The points that are connected by purple arrows denote inversion pairs

Fig. 2b–e. The BCs are presented along the $k_y$-direction across the X valley (the blue dashed line in Fig. 2a). As $P$ increases, two neighboring BC peaks with opposite signs develop and they gradually move away from each other (Fig. 2b). Once the ferroelectricity flips, the sign of the BC distribution also changes (Fig. 2b, c). In contrast, the SOC alters the BC insignificantly (Fig. 2d, e). The BC profile is well maintained regardless of the value of $\lambda$. The dependence on $P$ and $\lambda$ demonstrates that the BC is governed by the ferroelectricity rather than the SOC.

Based on the following symmetry argument, we can qualitatively interpret the behavior of the BC distributions with respect to ferroelectricity, which is summarized in Fig. 2f, g. We decompose the BC for spin $\sigma(=\pm)$ bands in terms of $\Omega_\sigma(\mathbf{k})$ as $\Omega(\mathbf{k}) = \Omega_+(\mathbf{k}) + \Omega_-(\mathbf{k})$. Under the time-reversal ($\mathcal{T}$) and mirror-reflection ($M_{xz}$) operations, $\Omega_\sigma(k)$ is transformed as follows

$$\Omega_\pm(\mathbf{k}) \xrightarrow{\mathcal{T}} -\Omega_\mp(-\mathbf{k}), \tag{2}$$

$$\Omega_\pm(k_x, k_y) \xrightarrow{M_{xz}} -\Omega_\mp(k_x, -k_y). \tag{3}$$

Eq. (3) follows from both the BC and the spin lying on the $xz$ mirror plane. As a result, $\Omega(\mathbf{k}) = -\Omega(-\mathbf{k})$ and $\Omega(k_x, k_y) = -\Omega(k_x, -k_y)$ due to the time-reversal symmetry and the mirror symmetry of the system, respectively. In the absence of $P$, the system recovers the inversion symmetry that leads to

$\Omega_\pm(\mathbf{k}) = \Omega_\pm(-\mathbf{k})$. Consequently, $\Omega(k)$ is zero without $P$ (Fig. 2f). If we consider the ferroelectric reversal, $\Omega_\pm^{+P}(\mathbf{k}) = \Omega_\pm^{-P}(-\mathbf{k})$ because the $+P$ and $-P$ configurations are inversion partners. Combining this with Eq. (2) yields $\Omega^{+P}(\mathbf{k}) = -\Omega^{-P}(\mathbf{k})$ (Fig. 2g). The symmetry argument well explains the DFT calculation results that reveal the ferroelectrically coupled BC.

**Origin of the ferroelectricity-coupled BC.** By extending the Hamiltonian of Eq. (1), we now construct an analytic model that provides microscopic origin on how the BC distribution can couple with the ferroelectricity. The ferroelectric polarization introduces new hopping channels that, when combined with the orbital spitting in Eq. (1), are the primary origin of the BC arising in the SnTe monolayer.

Figure 3a, b describes the new hopping channels driven by the ferroelectric displacement; an effective interorbital hopping channel from the Sn-$p_x$ orbital to the nearest-neighbor Sn-$p_y$ orbital along the $y$-axis is activated by the in-plane ferroelectricity. When the ferroelectric polarization has stabilized, the asymmetric hopping integrals are developed in Fig. 3a. After integrating out the Te atoms, an effective anti-symmetric interorbital hopping occurs with a hopping amplitude of $t_{xy}^{eff}$ that is proportional to the ferroelectric polarization. Upon the reversal of the ferroelectric polarization, the sign of the effective hopping is reversed (Fig. 3b).

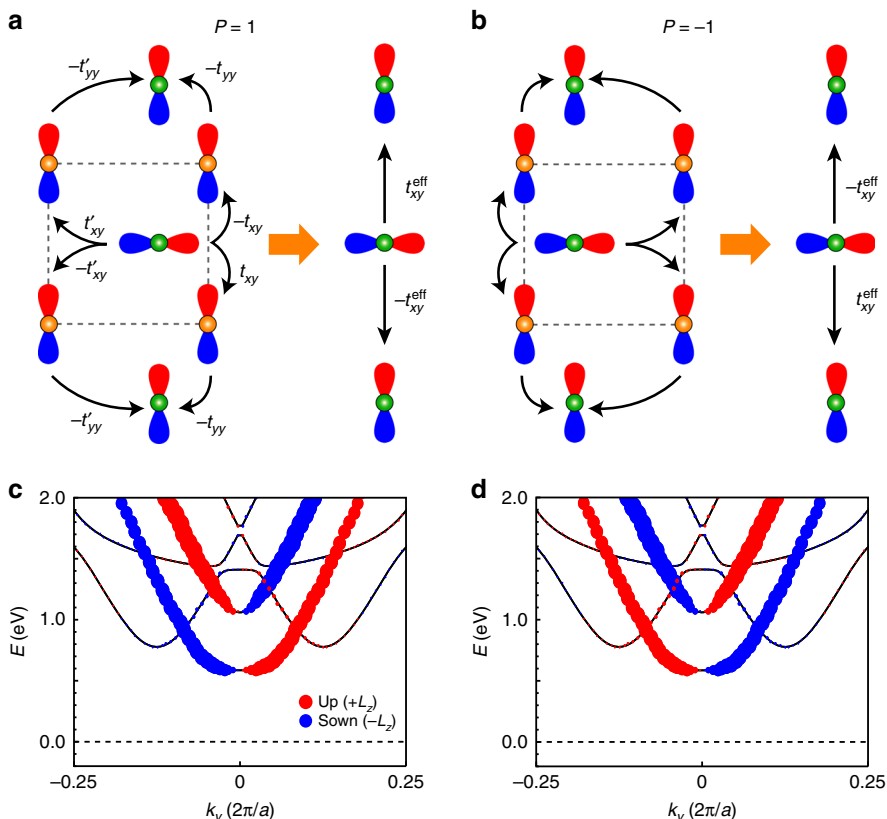

**Fig. 3 Ferroelectrically driven orbital Rashba effect. a, b** The emergence of an asymmetric inter-orbital hopping that is induced by a ferroelectric polarization, which results in the orbital Rashba effect. An effective inter-orbital hopping ($t_{xy}^{eff} = \frac{t_{xy}t_{yy} - t'_{xy}t'_{yy}}{E_{Sn} - E_{Te}}$) from the Sn-$p_x$ orbital to the nearest-neighbor Sn-$p_y$ orbital along the $y$-axis is allowed by the ferroelectric displacement. The sign of the effective hopping is determined by the polarization direction. The dumbbells represent $p_x$ and $p_y$ orbitals of Sn and Te atoms, whose atomic energies are $E_{Sn}$ and $E_{Te}$, respectively; the blue (red) colored region means a positive (negative) value of the $p$ orbital wave-function. Green balls, Sn; Orange balls, Te. **c, d** The orbital angular momentum texture that was obtained via density functional theory calculations in the absence of spin-orbit coupling depends on the polarization direction, which supports the presence of the ferroelectrically coupled orbital Rashba effect. As predicted in our analytic model, the orbital angular momentum is an odd function with respect to $k_y$ and the orbital-split-off states show opposite signs at the same $k$ point. From the close correlation between Berry curvature (BC) and orbital angular momentum, the orbital Rashba effect can bring a similar BC distribution, possibly leading to the BC dipole along the $k_y$-direction

Such a ferroelectrically driven hopping Hamiltonian for Sn can be expressed as $H_{FE}(\mathbf{k}) = it_{xy}^{eff}(|p_y\rangle\langle p_x| - |p_x\rangle\langle p_y|)\sin k_y a' \approx t_{xy}^{eff}a'k_y\tau_y$ near the X valley, where $a' = \sqrt{2}a$ is the distance between neighboring Sn atoms (see Supplementary Note 1 for a more rigorous tight-binding approach and its justification by comparison with DFT calculations). A similar Hamiltonian can be derived for electrons in Te atoms as well.

The ferroelectrically driven model Hamiltonian is equivalent to

$$H_{FE}(\mathbf{k}) = \alpha_L k_y L_z = |\alpha_L|\mathbf{L}\cdot(\hat{\mathbf{P}}\times\mathbf{k}), \qquad (4)$$

since the orbital angular momentum operator, $L_z$, is represented by $\hbar\tau_y$ in the $p_{x/y}$-orbital subspace. Here, $\alpha_L = t_{xy}^{eff}a'/\hbar$ is proportional to the ferroelectric polarization. We assumed that the direction of the ferroelectric polarization ($\hat{\mathbf{P}}$) is taken as $+\mathbf{x}$ without loss of generality. Eq. (4) is identical to the Rashba-type Hamiltonian[32,33] if we replace the spin angular momentum $\mathbf{S}$ by the orbital angular momentum $\mathbf{L}$; therefore, it can be referred to as the orbital Rashba effect[35]. Here, we explicitly demonstrate the emergence of the orbital Rashba effect from the ferroelectrically allowed interorbital hybridizations in Fig. 3a, b. In the absence of the ferroelectric polarization, such interorbital hopping channels are canceled out due to the inversion symmetry (see Supplementary Fig. 4). Consistently with the DFT calculations of Fig. 3c, d, the orbital Rashba effect produces the $z$-component orbital angular momentum texture that is an odd function of both $k_y$ and ferroelectric polarization. From the close correlation between BC and orbital angular momentum, as exemplified in graphene[8] and MoS2[9], one can expect that the orbital Rashba effect can bring a similar BC distribution.

To examine the relation between the ferroelectricity and the BC, we diagonalize the total Hamiltonian $H_0(\mathbf{k}) + H_{FE}(\mathbf{k})$ and obtain the following expression for the BC

$$\Omega(\mathbf{k}) = \frac{2\alpha_L J^2}{J_{\mathbf{k}}^3\hbar}\partial_{k_x}\theta_{\mathbf{k}}, \qquad (5)$$

where $J_{\mathbf{k}} = \sqrt{J^2 + (\alpha_L\hbar k_y)^2}$ is the modified orbital splitting (for details, see Methods and Supplementary Note 2). The linear $\alpha_L$-dependence of the BC implies that $\Omega(k)$ is induced by the ferroelectric polarization and it is switchable by reversing $P$. Since $\theta_{\mathbf{k}} = \arg(k_x + ik_y)$, the BC is an odd function of $k_y$, and thus it forms a dipole along the $y$-direction. This result indicates that the ferroelectrically driven orbital Rashba effect plays a central role in developing the BC dipole in the SnTe monolayer.

**Intra/interband BC dipoles and nonlinear responses**. The steep slope between the two adjacent and opposite BC peaks at the X valley gives rise to a large BC dipole, which induces nonlinear optoelectronic responses such as the nonlinear Hall effect and the circular photogalvanic effect[16–18]. By using the DFT method, we calculate the intraband BC ($\Omega_v$), interband BC ($\Omega_{vc}$) distributions, and their dipoles as shown in Fig. 4 (for the formalisms, see Methods). Due to the mirror symmetry $M_{xz}$, both the intraband and interband BC dipoles are composed of the $y$-component only. $\Omega_v$ and $\Omega_{vc}$ show similar distributions along the $k_y$-direction, leading to large BC dipoles. For a more accurate estimation, we performed our calculations with two exchange-correlation potentials, namely, the Perdew–Burke–Ernzerhof (PBE)[36] and Heyd–Scuseria–Ernzerho (HSE)[37] functionals, which yield the same qualitative trend.

When the SnTe monolayer is doped, the intra-band BC dipole $\mathbf{D}^{intra}(\mu)$ emerges and produces the nonlinear Hall current in response to a low-frequency photon[18]. By assuming that the chemical potential $\mu$ is close to the band edge $\mu_0$ of the parabolic band, we obtain $D_y^{intra}(\mu) = -\alpha_L(m/\pi J\hbar^3 k_X^2)|\mu - \mu_0|$ from Eq. (5). This induces a nonlinear Hall effect in the SnTe monolayer; a DC Hall current flows along the $x$-direction in response to the normal incident light with $y$-polarization and reveres its direction upon ferroelectric reversal.

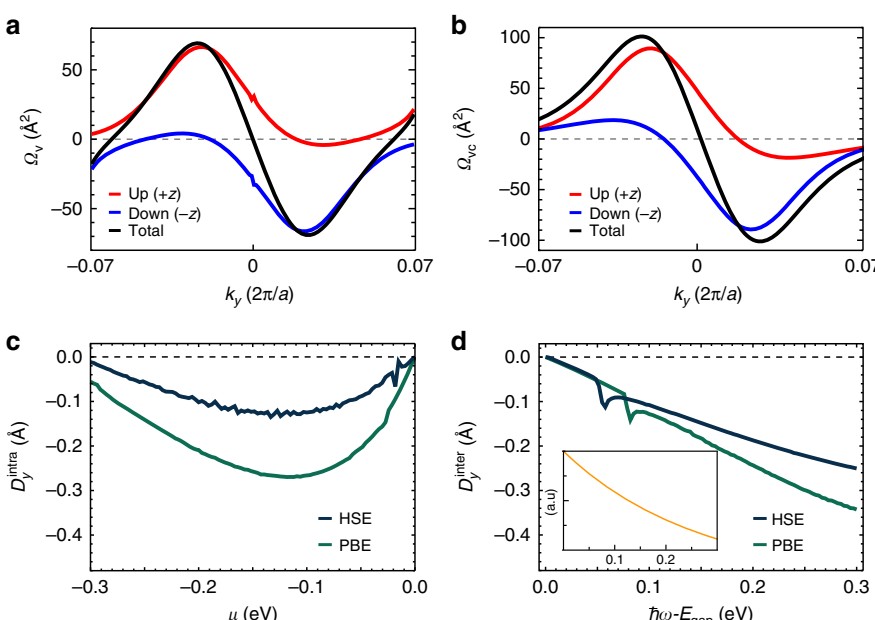

**Fig. 4** Berry curvature and Berry curvature dipoles. **a** The calculated Berry curvatures (BCs) for the highest valence bands near the X valley. The spinup (spindown) component is represented by a red (blue) line. **b** The calculated inter-band BCs between the highest valence bands and the lowest conduction bands near the X valley. The spinup (spindown) component is represented by a red (blue) line. **c** The intraband BC dipole as a function of the Fermi level and **d** the interband BC dipole as a function of the photon frequency, which were calculated from the PBE and HSE functionals. The inset shows the interband BC dipole calculated from our analytic formalism, Eq. (6), in which the unit of the horizontal axis is the same as the numerical result and the vertical axis of the interband BC dipole is presented in arbitrary unit

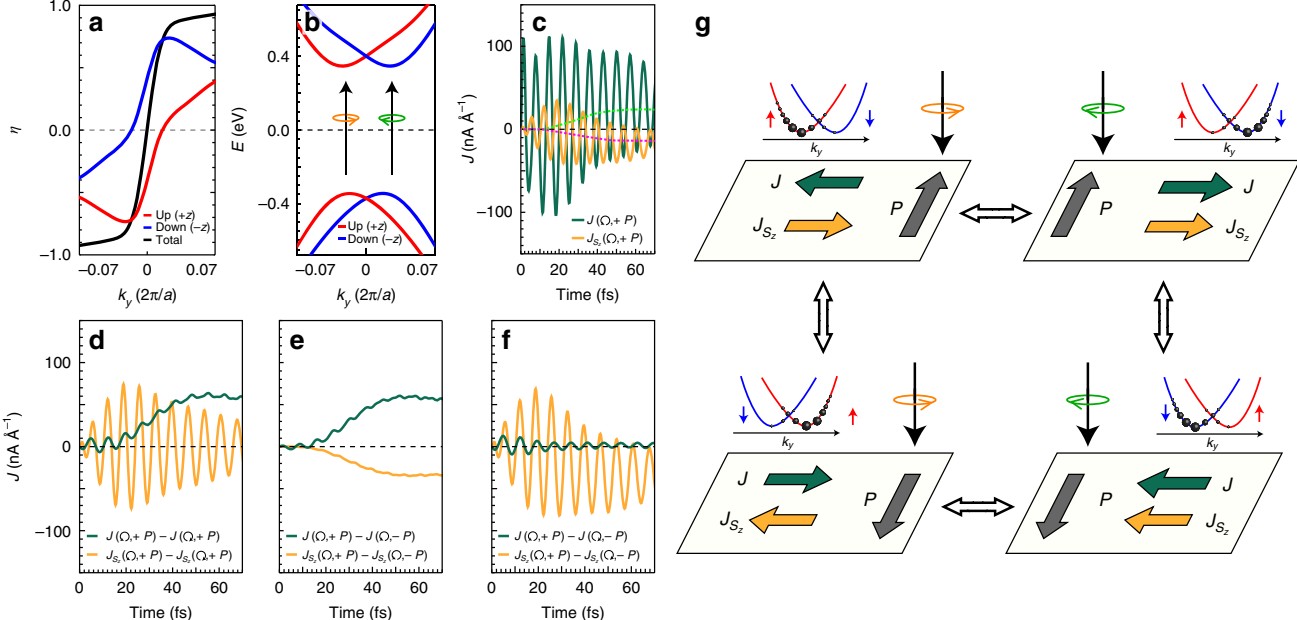

**Fig. 5** Switchable charge and spin photogalvanic effects. **a** The calculated circular dichroism ($\eta$) for the band-edge transitions near the X valley. The adsorption of the left-handed (right-handed) light prevails when $\eta$ is greater (smaller) than zero. **b** Spin-selective circular dichroic excitation near the X valley. The spinup (spindown) states are marked by red (blue) lines. An orange (green) arrow represents the right-handed (left-handed) light. **c** Charge (green line) and spin (yellow lies) currents under the right-handed light, which were calculated by the time-dependent density functional theory method. The dashed guidelines (cyan and magenta) were obtained by averaging out the rapidly oscillating contributions and represent the charge and spin photocurrents, respectively. **d–f** The calculated charge (green line) and spin (yellow lines) current differences that are caused **d** by the photon helicity, **e** by the ferroelectric polarization, and **f** by both the photon helicity and the ferroelectric polarization. Here, the spin current in (**d**) and charge current in (**f**) gives no contribution after averaging out the oscillating contributions. This observation implies that the reversal of the photon helicity changes the direction of the charge current only, while the reversal of the ferroelectric polarization reverses the direction of both charge and spin photocurrents. **g** Schematic drawings of the circular photogalvanic effect being controlled by the helicity of the circularly polarized light and the ferroelectric polarization direction; the observations in (**c–f**) for four different combinations of the photon helicity and the ferroelectric polarization are depicted. In this way, one can independently control the direction of the charge and spin currents induced by the incident circularly polarized light. Here, the inversion and time-reversal operations are denoted by the vertical and the horizontal white arrows, respectively

For the pristine insulating phase, the inter-band BC dipole $\mathbf{D}^{\text{inter}}(\omega)$ determines the inter-band circular photogalvanic effect[20]. Provided that the incident photon energy $\hbar\omega$ is comparable to the direct bandgap $E_{\text{gap}}$ at the X valley, optoelectronic responses are mainly governed by the band-edge transitions. Then, our analytic formalism gives the following expression for the inter-band BC dipole

$$D_y^{\text{inter}}(\omega) \propto \alpha_L \left(1 - \frac{E_{\text{gap}}}{\hbar\omega}\right), \qquad (6)$$

whose complete expression is presented in Supplementary Eq. (20). The inset of Fig. 4d presents the frequency dependence of Eq. (6), which accords with our DFT results. The circular photogalvanic current is then calculated from the relation[20] $J_{y,\pm} = \pm(2\pi e^3 \tau E_0^2 / \hbar^2) D_y^{\text{inter}}(\omega)$, where $\pm$ refers to the incident photon helicity, $\tau$ is the momentum relaxation time, and $E_0$ is the field amplitude of the light.

The order of magnitude of the BC dipoles in the SnTe monolayer (~0.1 Å) is comparable to that of the small-gap or gapless topological materials[20,21]. For instance, 0.1 Å of the interband BC dipole has been obtained in the WTe$_2$ monolayer under an external field of $E_z \sim 1.5$ V nm$^{-1}$, giving rise to ~200 nA W$^{-1}$ circular photogalvanic current[20]. Therefore, the SnTe monolayer is a promising platform for exploring the BC-related nonlinear optoelectronic responses over a wide frequency range.

## Discussion

When combined with the Rashba spin splitting that is caused by the ferroelectricity and the SOC, the unique BC structure yields additional fascinating optoelectronic responses in the SnTe monolayer; one can control the spin polarization of the photo-current as well as its charge degree of freedom. According to the spin-resolved BC profiles that are depicted in Fig. 4a, b, the positive and negative BC peaks are dominated by the spinup and spindown components, respectively. Such strong coupling of the spin polarization and the BC distribution leads to the spin- and momentum-asymmetric circular dichroism in Fig. 5a. Via the combination of the circular dichroism and the large Rashba spin splitting, each spin split-off band can be selectively excited by circularly polarized light with normal incidence[38], thereby producing a current-carrying nonequilibrium electron distribution (Fig. 5b). As a result, we can generate spin-polarized circular photogalvanic currents in the SnTe monolayer. Furthermore, such charge and spin photocurrents can be separately configured via circular dichroism and ferroelectric polarization.

Using the time-dependent DFT method to describe the nonequilibrium electron dynamics, we directly demonstrate the generation of charge and spin circular photogalvanic currents along with their possibility to be manipulated. When the SnTe monolayer is exposed to a time-varying circularly polarized electric field whose frequency is tuned to the band gap, the charge and spin currents are generated as shown in Fig. 5c. Although the time-dependent DFT method captures all optoelectronic/spin-tronic responses from first to higher order contributions, we

identify nonzero direct current (DC) components by plotting guidelines in Fig. 5c, which are related to the BC dipole. We confirmed that the DC component of the current is consistent with the second order response theory[39] (see Supplementary Fig. 5). These charge and spin currents flow perpendicular to the ferroelectric polarization and vary by switching the photon helicity and ferroelectric polarization.

When the photon helicity changes along with the fixed ferroelectric polarization, the charge current flows backwards while the spin current is unaltered, i.e., the difference between the spin currents under the right- and left-handed circularly polarized lights is zero (Fig. 5d). Upon varying the photon helicity, the spin and the group velocity of the excited carrier are simultaneously reversed, thus affecting the charge current only (see the parabolic bands with black dots in Fig. 5g). In Fig. 5e, the charge and spin current differences have finite values as the ferroelectric polarization is flipped while the photon helicity is fixed; hence, both the charge and spin currents reverse their directions. Under ferroelectric reversal, both the spin and the BC are flipped in the electronic structure. The crystal momentum of the excited carrier is then reversed while its spin direction is fixed, resulting in the reversal of both the charge and spin currents. If we change the ferroelectric polarization and the photon helicity at the same time, the spin current is reversed, whereas the charge current remains unaffected (Fig. 5f). Therefore, the charge and spin degrees of freedom in circular photogalvanic current can be readily controlled simultaneously and/or separately by means of the photon handedness and the ferroelectricity (Fig. 5g).

The switchable behavior of the charge and spin currents depicted in Fig. 5g can also be understood by the following symmetry argument. The two oppositely polarized configurations ($+P$ and $-P$) are connected by the inversion operation (vertical white arrows). And the charge and spin currents are odd under the spatial inversion. Therefore, both charge and spin photocurrents should be reversed under the ferroelectric reversal. On the other hand, the time-reversal operation transforms the SnTe monolayer exposed to the right circularly polarized light into the one to the left circularly polarized light, and vice versa. Under the time reversal (horizontal white arrows), the charge current is odd while the spin current is even. As a result, the reversal of the photon helicity inverts the direction of the charge current only, remaining the spin current invariant. In addition to the time-dependent DFT calculations and the symmetry argument, we develop an analytic formalism for the charge and spin photogalvanic currents to verify their ferroelectric origin and the maneuverability by the photon handedness and the ferroelectricity [see Supplementary Eqs. (21), (22), (24), and (25)].

There are two side remarks. First, our main results are likely to be valid for thicker SnTe films in which the in-plane ferroelectricity has been reported to be retained[27]. Moreover, the series of group-IV monochalcogenide monolayers share the same ferroelectrically driven BC-dipole features due to their similar electronic structures[31]. Second, the ferroelectrically driven BC dipole in the SnTe monolayer is different from the BC dipole proposed in the surface of the well-known topological crystalline insulator SnTe[18] where the singular BC distribution originates from the gapped and tilted surface Dirac cones. In our work, a sizeable BC dipole appears in a trivial and large insulating gap of the SnTe monolayer with the help of ferroelectricity.

In summary, we identified the fundamental relation between the ferroelectricity and the BC dipole, which is a counterpart of the well-known ferromagnetism and BC monopole coupling. Based on this finding, we demonstrated the possibility of generating and manipulating photocurrents via the ferroelectrically driven BC dipole in the SnTe monolayer. Despite the large gap in

the SnTe monolayer, its intraband and interband BC dipoles were predicted to reach substantial values that are comparable to those of the experimentally measured WTe$_2$ monolayer. The antisymmetric interorbital hopping that is induced by the ferroelectricity gives rise to the orbital Rashba effect, which plays an essential role in the BC dipole structure. In addition, we presented the charge and spin circular photogalvanic currents and, on the basis of the light handedness and ferroelectric polarization switching, we proposed a pragmatic scheme for simultaneously or independently controlling them. Through the large ferroelectrically driven and, thus, ferroelectrically controlled BC dipole, the SnTe monolayer can serve as a unique platform for engineering the BC in a non-volatile way and has high potential for optoelectronic and optospintronic applications.

## Methods

**Electronic structure calculation.** Our DFT calculations are performed using the projected augmented plane-wave method[40,41] as implemented in the Vienna ab initio simulation package[42]. The optimized atomic structure of the SnTe monolayer is obtained from the HSE functional[37]. The PBE functional of the generalized gradient approximation is used to describe the exchange–correlation interactions among electrons[36]. The isolated SnTe monolayer is considered within supercell geometries where the interlayer distance is 15 Å in the surface normal direction. The energy cutoff for the plane-wave-basis expansion is selected to be 450 eV. We used a $10 \times 10 \times 1$ **k**-point grid to sample the entire Brillouin zone. The BC $\Omega(k)$ is calculated as follows[1,43]

$$\Omega(\mathbf{k}) = -2\mathrm{Im} \sum_n \sum_{n' \neq n} f_n \frac{\langle \psi_n(\mathbf{k}) | v_x | \psi_{n'}(\mathbf{k}) \rangle \langle \psi_{n'}(\mathbf{k}) | v_y | \psi_n(\mathbf{k}) \rangle}{(E_{n'}(\mathbf{k}) - E_n(\mathbf{k}))^2}, \quad (7)$$

where $n$ is the band index, $f_n$ is the Fermi–Dirac distribution function, $v_{x(y)}$ is the velocity operator, and $\psi_n(\mathbf{k})$ and $E_n(\mathbf{k})$ are the Bloch wave-function and energy, respectively, of the $n$th band at point **k**. The BC and the spin BC are evaluated via the maximally localized Wannier function using the WANNIER90 package[44,45]. The intraband BC and interband BC dipoles are estimated using a $2000 \times 2000 \times 1$ **k**-point grid.

**Intra/interband BC dipoles.** The intra-band BC dipole, namely, **D**$^{\mathrm{intra}}$, is expressed as a function of the chemical potential $\mu$ as follows

$$\mathbf{D}^{\mathrm{intra}}(\mu) = \frac{1}{(2\pi)^2} \sum_n \int f(E_n(\mathbf{k}) - \mu) \nabla_{\mathbf{k}} \Omega_n(\mathbf{k}) d^2 k, \quad (8)$$

where $\Omega_n$ represents the $n$th band BC and $f(E_n(\mathbf{k}) - \mu)$ is the Fermi–Dirac distribution. The interband BC dipole **D**$^{\mathrm{inter}}$ is expressed as

$$\mathbf{D}^{\mathrm{inter}}(\omega) = \frac{1}{(2\pi)^2} \int \Theta(\hbar\omega - \Delta E(\mathbf{k})) \nabla_{\mathbf{k}} \Omega_{\mathrm{vc}}(\mathbf{k}) d^2 k. \quad (9)$$

Here, $\Omega_{\mathrm{vc}} = i \sum_{v,c} \langle v | \partial_{k_x} H | c \rangle \langle c | \partial_{k_y} H | v \rangle / [\Delta E(\mathbf{k})]^2$ is the interband BC[20] between the valence and conduction bands [the summation running over each spin band of the valance ($v$) and the conduction ($c$) band], $\Delta E(\mathbf{k})$ is the interband transition energy between them, $\hbar\omega$ denotes the photon energy, and $\Theta$ is the Heaviside step function.

**Charge and spin photogalvanic current calculation.** To estimate the charge and spin currents induced by the circularly polarized light, the time-dependent DFT calculations are performed using a custom code based on the quantum ESPRESSO package[46,47]. The exchange and correlation interactions between electrons are described by the PBE-type generalized gradient approximation functional[36]. Norm-conserving pseudopotentials are used to describe the nuclei-electron interaction. In addition to the $10 \times 10 \times 1$ grid points that are sampled by the Monkhorst–Pack scheme in the Brillouin zone, 16 time-reversal-symmetric **k**-points near the X valleys are employed to simulate the photo-excited currents more accurately. To investigate the real-time dynamics of the optical responses, circularly polarized light is applied in the velocity gauge $\mathbf{A}_{\pm}(t) = \frac{E_0}{\omega_0}(\mathbf{x} \sin(\omega_0 t) \pm \mathbf{y} \cos(\omega_0 t))$, where the electric field amplitude is $E_0 = 5.64 \times 10^{-3}$ V Å$^{-1}$, and the light frequency is set to the resonant direct band gap energy as $\hbar\omega_0 = 0.58$ eV. The charge and spin currents induced by the circularly polarized light are evaluated in terms of the expectation values of the spin and velocity operators as follows[48]

$$\mathbf{J}(t) = \frac{-e}{V_2} \sum_{\mathbf{k}}^{\mathrm{BZ}} \sum_n f_n \langle \psi_n(\mathbf{k}, t) | \mathbf{v} | \psi_n(\mathbf{k}, t) \rangle, \quad (10)$$

$$\mathbf{J}_{S_z}(t) = \frac{e}{2V_2} \sum_{\mathbf{k}}^{\mathrm{BZ}} \sum_n f_n \langle \psi_n(\mathbf{k}, t) | \{S_z, \mathbf{v}\} | \psi_n(\mathbf{k}, t) \rangle. \quad (11)$$

where $n$ is the band index, $f_n$ is the initial occupation of the Bloch state, and $V_2$ is the lattice surface area.

## Data availability
The data that support the findings of this study are available from J.K. and K.W.K. upon reasonable request.

## Code availability
The code that was used for the time-dependent DFT calculations is available from N.P. upon reasonable request.

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

## Acknowledgements

Financial support from the Basic Science Research Program of the National Research Foundation of Korea (NRF) under Grant nos. 2016R1D1A1B03933255, 2017M3D1A1040828, 2019R1A2C1010498 (H.J.), and 2016R1D1A1B03931542, 2017R1A4A1015323 (D.S. and N.P.) is gratefully acknowledged. J.K. was supported by the NRF grant funded by the Korea government (MSIT) (No. 2019R1F1A1059743) and by Incheon National University Research Grant in 2019 (20190291). K.W.K. acknowledges financial support from the KIST Institutional Program and the National Research Council of Science & Technology (NST) (Grant no. CAP-16-01-KIST). K.W.K. and J.S. were also supported by the German Research Foundation (DFG) (No. SI 1720/2-1), the Alexander von Humboldt Foundation, and the Transregional Collaborative Research Center (SFB/TRR) 173 SPIN+X.

## Author contributions

J.K. and S.H.L. performed first-principles calculations. D.S. and N.P. conducted the time-dependent DFT calculations. K.W.K. and J.S. constructed the model Hamiltonian and calculated the corresponding results. J.K., K.W.K. and H.J. analysed the data and wrote the paper with the help of the other authors.

## Additional information

**Competing interests:** The authors declare no competing interests.

