## [Peer Review File · Nature Communications]

Reviewers' comments:

Reviewer #1 (Remarks to the Author):

This manuscript reports a novel ferroelectricity driven Berry curvature dipole in SnTe monolayer for the first time, and discusses its implication on charge, spin, and its optoelectronic response. The work appears very carefully executed and results thoroughly analyzed, both quantitatively and qualitatively through symmetry arguments. Since ferroelectricity can serve as a non-volatile state, it enables an interesting avenue to encode Berry curvature information in the form of electric polarization. Certainly, the work is novel and I would strongly recommend its publication in Nature Communications. Several comments are,

1. English needs proof reading. I came across a number of sentences with awkward phrasing e.g. "Berry curvature (BC) has become more important due to its central role..." might be better phrased as "The concept of Berry curvature (BC) is becoming increasingly pertinent due to its central role..."
2. The authors state in the first paragraph of the main text, "Under out-of-equilibrium electron distributions through intra-or inter-band excitations, the BC dipole can allow the nonlinear optoelectronic transport..." My understanding is that BC is due only to interband processes. Please clarify.
3. When first referencing to BC dipole, the authors should also cite this early work, <https://journals.aps.org/prb/abstract/10.1103/PhysRevB.92.235447>
4. The authors said "The ferroelectric polarization breaks the inversion symmetry". I might have been mistaken here, but I thought monolayer buckled phase SnTe, which has similar structure as black phosphorus, has broken inversion symmetry. Is the ferroelectricity mainly enhancing the BC?
5. Does going from monolayer to bilayer restore the inversion symmetry and eliminate the BC? Please discuss.

Reviewer #2 (Remarks to the Author):

The manuscript reviews on presenting "Ferroelectric-driven Berry curvature dipole with switchable charge and spin circular photogalvanic effect in SnTe monolayer". The authors propose an idea that ferroelectricity could rise a pair of positive and negative Berry curvature peaks and further Berry curvature dipoles in a large-gap system. An example of SnTe is taken as a platform to explore the Berry curvature related nonlinear optoelectronic responses. Here are my comments:

1. The current manuscript does not meet the high publication standards and style of Nature Communications. The authors list quite a lot results, however, some of them are not relevant to Berry curvature dipole, such as Fig. 1c&d, which are confusing because they seem irrelevant details.
2. Spin Berry curvature mentioned in the manuscript seems also not the relevant topic. As we can see in Fig. 2, when there is without ferroelectricity, non-zero spin Berry curvature distribution still exists. I am confused why spin Berry curvature is mentioned here, does it influence Berry curvature dipole or is it relevant to spin circular photo galvanic effect?

3. Since the sign of the ferroelectric polarization influence the sign of responses, I suggest that the results of negative P should be added in Fig. 2(b,c,e,f), or there should plot a phase diagram.

4. In the last equation of Page15 of manuscript, spin velocity operator should be written as $\{S_z, v\}/2$.

5. It is better for the author to calculate the charge and spin circular photogalvanic effect based on the constructed effective model (Eqs. 2(a)-2(d)) using the corresponding formula of the circular photogalvanic effect (similar to Fig.4), which can be compared with the results directly calculated by TDDFT, thus verifying the conclusion of this paper.

I suggest that the manuscript should be better-organized and some details should be placed in Supplemental Information, then submit it again.

Reviewer #3 (Remarks to the Author):

The manuscript by Jeongwoo Kim et al. presented theoretical studies of the ferroelectricity and Berry curvature properties in the monolayer SnTe. The manuscript is well-written. The authors communicated the broad significance of the topic and the specific approach for their work very well. The symmetry analyses are corrected and well done. The DFT theoretical calculations and tight-binding modellings are both extensive and systematic. Therefore, I support this paper for publication.

Two minor suggestions

1. The authors may think about also computing the in-plane ferroelectric polarization using the Berry phase theory.
2. The authors may think about computing the photocurrents as a function of photon energy.

We would like to thank all the referees for their invaluable criticism and comments. We have revised our manuscript by adding more explanations based on referees' comments. Here we make a list of our responses to the referee comments, by quoting the original comment, and changes made accordingly.

Reply to Reviewer #1

This manuscript reports a novel ferroelectricity driven Berry curvature dipole in SnTe monolayer for the first time, and discusses its implication on charge, spin, and its optoelectronic response. The work appears very carefully executed and results thoroughly analyzed, both quantitatively and qualitatively through symmetry arguments. Since ferroelectricity can serve as a non-volatile state, it enables an interesting avenue to encode Berry curvature information in the form of electric polarization. Certainly, the work is novel and I would strongly recommend its publication in *Nature Communications*. Several comments are,

We are very pleased with the positive comments and suggestions for improving our manuscript, which we deeply appreciate.

1. English needs proof reading. I came across a number of sentences with awkward phrasing e.g. "Berry curvature (BC) has become more important due to its central role..." might be better phrased as "The concept of Berry curvature (BC) is becoming increasingly pertinent due to its central role..."

(ANS) Following the referee's advice, our manuscript has been carefully modified with the aid of a professional English editing service. We believe that the integrity of the manuscript has been considerably improved to the extent which *Nature Communications* requires.

2. The authors state in the first paragraph of the main text, "Under out-of-equilibrium electron distributions through intra- or inter-band excitations, the BC dipole can allow the nonlinear optoelectronic transport..." My understanding is that BC is due only to inter-band processes. Please clarify.

(ANS) We intended to describe general features of the nonlinear dynamics which can be affected by intra- and inter-band excitations combined with the BC distribution encoded in the band structures [PRL 115, 216802 (2015)]. However, we agree with the referee that the expression can be misleading by making an unwanted connection between the intra-band excitation and the BC distributions. Thus, in this revision, we remove "through intra- or inter-band excitations" to avoid the ambiguity.

3. When first referencing to BC dipole, the authors should also cite this early work,

<https://journals.aps.org/prb/abstract/10.1103/PhysRevB.92.235447>

(ANS) We thank the referee for bringing to our attention the important work that should have been included in the previous version. Following the referee's suggestion, we cite the work in the revised version (Ref. 19).

4. The authors said "The ferroelectric polarization breaks the inversion symmetry". I might have been mistaken here, but I thought monolayer buckled phase SnTe, which has similar structure as black phosphorus, has broken inversion symmetry. Is the ferroelectricity mainly enhancing the BC?

(ANS) As pointed out by the referee, the SnTe monolayer has the similar crystal structure as the puckered structure of phosphorene, that is, the monolayer of black phosphorus. The crucial difference is that phosphorene has the inversion symmetry, but the SnTe monolayer does not. The inversion center of phosphorene occurs at the center of two neighboring phosphorus atoms located upper and lower sub-layer. As described in the main manuscript, the SnTe monolayer is the binary version of phosphorene, and hence the inversion center connecting the two inversion partner P atoms is not preserved in SnTe. As already mentioned in the main text, Sn and Te atoms show the opposite displacement along the [100] direction, inducing the in-plane ferroelectricity. (By contrast, in phosphorene, two P atoms moves oppositely, but there is no ferroelectricity.) If the ferroelectric displacement goes to zero, the SnTe monolayer recovers the inversion symmetry, and it becomes a (001) thin film of the rock-salt type square lattice. Therefore, the ferroelectricity is the single factor to break the inversion symmetry of the system and to enhance the large and switchable BC dipole, resulting in optoelectronic responses as we elaborated in the manuscript.

5. Does going from monolayer to bilayer restore the inversion symmetry and eliminate the BC? Please discuss.

(ANS) According to [Science 353, 274 (2016)], a SnTe bilayer also retains the in-plane ferroelectricity at room temperature, implying that the inversion symmetry is *not* restored. However, for other group-IV monochalcogenide bilayer systems, two neighboring monolayers show antiferroelectric ordering, and they restore the inversion symmetry, eliminating the BC as pointed out by the referee. We added the related discussion in the revised version (the 11th - 14th lines on page 12).

Reply to Reviewer #2

The manuscript reviews on presenting “Ferroelectric-driven Berry curvature dipole with switchable charge and spin circular photogalvanic effect in SnTe monolayer”. The authors propose an idea that ferroelectricity could rise a pair of positive and negative Berry curvature peaks and further Berry curvature dipoles in a large-gap system. An example of SnTe is taken as a platform to explore the Berry curvature related nonlinear optoelectronic responses. Here are my comments:

We thank the referee for reviewing our manuscript and making valuable suggestions strengthening our manuscript. Here, we addressed all the issues raised by the referee.

1. The current manuscript does not meet the high publication standards and style of Nature Communications. The authors list quite a lot results, however, some of them are not relevant to Berry curvature dipole, such as Fig. 1c&d, which are confusing because they seem irrelevant details.

2. Spin Berry curvature mentioned in the manuscript seems also not the relevant topic. As we can see in Fig. 2, when there is without ferroelectricity, non-zero spin Berry curvature distribution still exists. I am confused why spin Berry curvature is mentioned here, does it influence Berry curvature dipole or is it relevant to spin circular photo galvanic effect?

(ANS for #1 and #2) We agree with the referee that some part of our work (including Fig. 1c & 1d and the spin Berry curvature shown in the previous manuscript) are not straightforwardly associated with our main conclusion, *i.e.*, the ferroelectrically driven Berry curvature dipole and the switchable charge and spin photogalvanic effect, although they are, to some extent, useful for constructing and validating our model Hamiltonian. Following the referee’s suggestion, we re-drew Figs. 1 and 2 by removing the momentum-dependent orbital splittings and the spin Berry curvatures, respectively. We largely removed the descriptions on the spin Berry curvature and revised the manuscript in a convergent manner; please see the sections **Berry curvature and its dipole**, **Ferroelectrically driven orbital Rashba effect**, and **Comparison to DFT calculations** in the main text. The deleted information is now placed in the Supplementary Information and Extended Data Fig. 2 for motivated readers who seek the complete details of our study (the 9th - 11th lines on page 9).

3. Since the sign of the ferroelectric polarization influence the sign of responses, I suggest that the results of negative P should be added in Fig. 2(b,c,e,f), or there should plot a phase diagram.

(ANS) Following the referee's suggestion, the results of negative P are included in the revised Fig. 2 b-e. In addition, the BC distribution by reversing the ferroelectric polarization is depicted in Extended Data Fig. 3.

4. In the last equation of Page15 of manuscript, spin velocity operator should be written as $\{S_z, v\}/2$.

(ANS) We thank the referee for leaving an important comment. We changed the spin velocity operator in the equation. Since each Bloch state is diagonal in S_z operator, which is imposed by the glide mirror symmetry G , any results presented in the main text are not altered.

5. It is better for the author to calculate the charge and spin circular photogalvanic effect based on the constructed effective model (Eqs. 2(a)-2(d)) using the corresponding formula of the circular photogalvanic effect (similar to Fig.4), which can be compared with the results directly calculated by TDDFT, thus verifying the conclusion of this paper.

(ANS) We thank the referee for making this suggestion that can substantially improve our work. Following the referee's suggestion, we calculated the Berry curvature dipole and the charge/spin photocurrents by using our analytical formalism. With our tight-binding model, we successfully reproduced the main features presented in Figs. 4d and 5g. We briefly mention it in the revised manuscript (the 13th - 14th lines on page 10, the 9th - 10th lines on page 12) and present the detailed calculations in Supplementary Information.

I suggest that the manuscript should be better-organized and some details should be placed in Supplemental Information, then submit it again.

We again appreciate the referee's consideration of our manuscript as a potential candidate for publication in *Nature Communications* and his/her valuable suggestions reinforcing the manuscript. Based on the improvement, we are convinced that the revised manuscript is ready for publication in *Nature Communications*.

Reply to Reviewer #3

The manuscript by Jeongwoo Kim et al. presented theoretical studies of the ferroelectricity and Berry curvature properties in the monolayer SnTe. The manuscript is well-written. The authors communicated the broad significance of the topic and the specific approach for their work very well. The symmetry analyses are corrected and well done. The DFT theoretical calculations and tight-binding modellings are both extensive and systematic. Therefore, I support this paper for publication.

We thank the referee for reviewing our manuscript and supporting publication of our manuscript. We reflected the referee's two suggestions in the revised manuscript as below.

Two minor suggestions

1. The authors may think about also computing the in-plane ferroelectric polarization using the Berry phase theory.
2. The authors may think about computing the photocurrents as a function of photon energy.

(ANS for #1 and #2) Following the referee's suggestions, we computed the in-plane ferroelectric polarization based on the Berry phase theory, and the photocurrent as a function of photon energy. The results are presented in Extended Data Figs. 1 and 5 together with corresponding sentences in the main text (the 24th line on page 3 – the 1st line on page 4, the 13th – 14th lines on page 10).

We thank all the referees for these suggestions which have strengthened our manuscript. We thus believe that the revised manuscript is now suitable for publication in *Nature Communications*.

Reviewers' comments:

Reviewer #1 (Remarks to the Author):

The authors has satisfactory addressed all my comments. I do not have additional suggestions, and would recommend its publication in Nature Comm.

Reviewer #2 (Remarks to the Author):

In the revised manuscript, the authors have addressed some of my questions. Here are my comments:

1. The manuscript seems still not well-organized. The results are just listed in several sections without enough explanation to tell us their relationship. For example, the authors don't give a clear explanation about the relationship between Berry curvature dipole and charge/spin photo galvanic effect. In the current version of supplementary information, the authors provide analytic results of charge and spin current directly calculated with berry dipole, which is in line with Figure 5(g). I think this is the points should be mentioned in the main text to further clarify their relationship.

2. TDDFT results (Figure 5) include contribution from first order effect and second order effect, while berry curvature dipole is a second order effect. The authors should make a clear description that which character is relevant to berry curvature dipole.

3. The model's results of berry curvature dipole can put on the same figure with DFT in Fig.4.

4. Many parts of the manuscript does not bring us novel insights. The spontaneously ferroelectric transition in low temperatures of SnTe monolayer has been reported several years ago (Ref. 27 in the main text). In Ref. 18 of the main text, this spontaneously ferroelectric transition induces nonzero Berry curvature dipole in [001] surface of SnTe are reported. The only difference is that this manuscript focus on monolayer SnTe. Since the in-plane ferroelectric polarization, there is no essential difference. In addition, the berry dipole related nonlinear response are given in Ref.18. In brief, this version of the manuscript presents too many details on the tight binding analysis, which makes the manuscript away from the point.

5. The title is named as "Ferroelectrically driven Berry curvature dipole with switchable charge and spin circular photogalvanic effect in the SnTe monolayer", but the switchable photogalvanic effect is placed in discussion section without serious discussion.

6. The section "Berry curvature and its dipole" does not discuss anything relevant to "its dipole", and there is another section named "Intra/inter-band Berry curvature dipoles and nonlinear responses".

7. In the section of "Intra/inter-band Berry curvature dipoles and nonlinear responses", there is no discussion about nonlinear responses presented. The formula of Intra/inter-band Berry curvature dipoles are published result, it should be placed in Method.

8. Since SnTe is monolayer in the manuscript, "a van der Waals monolayer of SnTe" is not appropriate here.

In summary, this version of manuscript still lacks enough novelty and does not meet the high publication standards and style of Nature Communications, since the main physics is presented in the experimental work (Ref. 20 in the revised manuscript). A major revision with re-setting-up the logical procedure among sections is needed and the points should be clarified with enough

explanation. I would not recommend the publication of the manuscript.

Reviewer #3 (Remarks to the Author):

The authors have very thoroughly addressed my concerns. I therefore recommend the publication in Nature Communications.

We would like to thank the referee again for spending time in the review process and providing valuable suggestions. His/her helpful comments inspired us to think the related issues more deeply and strengthen the logical flow by making a substantial revision on the manuscript. Here, we present our responses to all the questions raised by the referee.

Reply to Reviewer #2

1. The manuscript seems still not well-organized. The results are just listed in several sections without enough explanation to tell us their relationship. For example, the authors don't give a clear explanation about the relationship between Berry curvature dipole and charge/spin photo galvanic effect. In the current version of supplementary information, the authors provide analytic results of charge and spin current directly calculated with berry dipole, which is in line with Figure 5(g). I think this is the points should be mentioned in the main text to further clarify their relationship.

(ANS) While we have tried to highlight the underlying physics and the intriguing photogalvanic phenomena, it seems that the relation among multiple physical quantities has not been clarified enough to convince the referee. Following the referee's suggestion, the analytic model part was largely rewritten. By simplifying the model construction, relocating several paragraphs, and removing relatively less important parts, we established the direct connection between the orbital Rashba effect and the Berry curvature distribution more clearly (the 12th – 21st lines of page 4, the 9th line of page 7 – 8th line of page 8). In "Intra/inter-band Berry curvature dipoles and nonlinear responses" section, we explicitly addressed analytic formalisms and their interpretations regarding the relationship between inter/intra-band Berry curvature dipoles and circular photogalvanic currents (the 20th line of page 8 – 13th line of page 9). From the model analysis, we showed that the Berry curvature, its dipole, and charge/spin photogalvanic currents are all linearly dependent on the ferroelectrically driven orbital Rashba coefficient α_L , which indeed confirms the ferroelectric origin of these quantities. Furthermore, we made symmetry arguments on the charge/spin photogalvanic currents which provide a general explanation on their switchable behavior (the 15th line of page 11 – 2nd line of page 12). Based on these revision, now we can safely claim that the interesting Berry-curvature-related phenomena in the SnTe monolayer are governed by the ferroelectricity. All the changes were highlighted in the revised manuscript.

2. TDDFT results (Figure 5) include contribution from first order effect and second order effect, while berry curvature dipole is a second order effect. The authors should make a clear description that which

character is relevant to berry curvature dipole.

(ANS) As the referee pointed out, the first order and the second order effects are simultaneously captured in TDDFT calculations. While the rapidly oscillating responses in Fig. 5c-f are mostly enforced by the first order effect, the nonvanishing DC components of photocurrents are generated by the second order effect via the Berry curvature dipole. We have cross-checked that the photocurrents of the SnTe monolayer are produced by the second-order optical response in DFT calculations (Extended Data Figure 5). In order to avoid ambiguities, we identified the DC photocurrents by plotting guidelines in Fig. 5c and described the clear connection between the Berry curvature dipole and photocurrents (the 17th – 21st lines of page 10).

3. The model's results of berry curvature dipole can put on the same figure with DFT in Fig.4.

(ANS) Following the referee's suggestion, we added the Berry curvature dipole plot obtained from the analytic model in the inset of Fig. 4d.

4. Many parts of the manuscript do not bring us novel insights. The spontaneously ferroelectric transition in low temperatures of SnTe monolayer has been reported several years ago (Ref. 27 in the main text). In Ref. 18 of the main text, this spontaneously ferroelectric transition induces nonzero Berry curvature dipole in [001] surface of SnTe are reported. The only difference is that this manuscript focus on monolayer SnTe. Since the in-plane ferroelectric polarization, there is no essential difference. In addition, the berry dipole related nonlinear response are given in Ref.18. In brief, this version of the manuscript presents too many details on the tight binding analysis, which makes the manuscript away from the point.

(ANS) We believe that the impact and novelty of our study lies in the facts that

- i) by revealing the microscopic origin of the direct coupling between ferroelectricity and BC dipole, we suggested the ferroelectricity as a new mechanism to provide a large BC dipole *even without* singular band-crossing or band-inversion,
- ii) through this direct coupling, we demonstrated the generation of charge and spin circular photogalvanic currents along with their rich manipulability.

More specifically, we want to address the significance of our work distinct from Ref. 27 and Ref. 18.

The presence of the ferroelectricity in SnTe thin films has been discovered in Ref. 27. On the other hand, the focus of our study is the microscopic origin of the unusual quantum geometrical phase

caused by the in-plane ferroelectricity rather than the ferroelectricity itself.

Ref. 18 has manifested the connection between the BC dipole and the nonlinear Hall effect by using the semi-classical Boltzmann equation, and proposed the tilted Dirac or Weyl semimetals as an important candidate system to generate a large BC dipole. Since the three dimensional bulk SnTe is the well-known topological crystalline insulator, its [001] surface hosts multiple massless Dirac points as topological surface states. When combined with ferroelectricity, half of the Dirac points at the [001] surface of SnTe become massive and tilted by the broken mirror symmetry, possibly resulting in a non-vanishing large BC dipole. Importantly, the ferroelectricity just plays a simple role as a symmetry breaking perturbation to make a tilted Dirac cones in Ref. 18, and thus the correlation between ferroelectricity and Berry curvature dipole, which is the main finding in this study, has not been identified at all. Due to the lack of this significant correlation, the switching behavior of the BC dipole under the ferroelectric reversal has not been found. Therefore, one of the key concepts of our study—the inter-band BC dipole and its switchable photogalvanic responses—has not been examined at all in Ref. 18. Consequently, the electronic and BC structures of the tilted Dirac cones that appear at the surface of bulk SnTe are different from those of the SnTe monolayer investigated in our study.

We would like to emphasize that we newly found the large Berry dipole generated by ferroelectricity even in a large-gap system, whereas previous studies including Ref. 18 mostly focused on gapless or small-gapped topological systems to utilize the singular BC induced by band-crossing or band-inversion. As we described in the manuscript, we revealed the general relation between BC dipole and ferroelectricity, which is the corresponding counterpart of the well-known coupling of BC monopole and ferromagnetism. We also demonstrated that the charge/spin photocurrents originating from the inter-band BC dipole (and the Rashba spin splitting) can be controlled in various ways via the ferroelectricity-BC coupling. Although the SnTe monolayer is mainly investigated as a case study in this work, the fundamental physics, the ferroelectricity-coupled BC, can be extended not only to the group-IV monochalcogenides monolayers showing the same in-plane ferroelectricity, but also to other bulk ferroelectric systems.

To address the referee's comment, in the revised version, we highlighted our original contributions more explicitly (the 6th – 11th lines of page 12). Moreover, we substantially rearranged the model description to make it more compact and concise in consistent with the numerical calculations (the 12th – 21st lines of page 4, the 9th line of page 7 – 8th line of page 8). The results not directly related to the BC dipole were relegated to the supplementary information. With the help of the referee's invaluable criticism, we believe we can successfully present the novelty and originality of our work in the revised version.

5. The title is named as “Ferroelectrically driven Berry curvature dipole with switchable charge and spin circular photogalvanic effect in the SnTe monolayer”, but the switchable photogalvanic effect is placed in discussion section without serious discussion.

(ANS) Inspired by the referee’s comment, we strengthened our argument on the switchability of the photocurrent in the SnTe monolayer by adding more formalisms from our analytic model and providing symmetry argument (the 24th line of page 10 – 14th line of page 11, the 15th line of page 11 – 2nd line of page 12). In the revised discussion part, the switchability of the charge/spin current were addressed in view of three distinct aspects; *i.e.*, the direct evaluation through TDDFT, the analytic model analysis, and the symmetry analysis. The complementary viewpoints provide profound insights on how the Berry curvature dipoles and the corresponding charge/spin photogalvanic currents are entangled with the spontaneous polarization.

6. The section “Berry curvature and its dipole” does not discuss anything relevant to “its dipole”, and there is another section named “Intra/inter-band Berry curvature dipoles and nonlinear responses”.

(ANS) We repeatedly appreciate referee's careful attention to our work. Following the referee’s comment, we changed the section title from “Berry curvature and its dipole” to “Ferroelectrically driven Berry curvature” to convey the correct information.

7. In the section of “Intra/inter-band Berry curvature dipoles and nonlinear responses”, there is no discussion about nonlinear responses presented. The formula of intra/inter-band Berry curvature dipoles are published result, it should be placed in Method.

(ANS) We agree that the details of the nonlinear responses caused by the Berry curvature dipole are insufficient in the previous version. In the revised manuscript, we added detailed explanations on how the nonlinear transport properties are triggered by linearly or circularly polarized light (the 20th line of page 8 – 13th line of page 9). The formulas of intra/inter-band Berry curvature dipole in Eq. (6) and (7) that had been written in the Results section were moved to the Method section.

8. Since SnTe is monolayer in the manuscript, “a van der Waals monolayer of SnTe” is not appropriate here.

(ANS) Following the referee's comment, we changed "a van der Waals monolayer of SnTe" to "a SnTe monolayer" in the revised manuscript.

In summary, this version of manuscript still lacks enough novelty and does not meet the high publication standards and style of Nature Communications, since the main physics is presented in the experimental work (Ref. 20 in the revised manuscript). A major revision with re-setting-up the logical procedure among sections is needed and the points should be clarified with enough explanation. I would not recommend the publication of the manuscript.

(ANS) We again appreciate the referee's careful reading and valuable suggestions. Although Ref. 20 successfully showed that the photogalvanic effect is induced by the Berry curvature dipole in the WTe₂ monolayer with the aid of external gate voltage, the large Berry curvature dipole still occurs in the tilted and small-gapped Weyl point, which was basically the same concept suggested in Ref. 18. It also has a limitation of the low frequency operation due to the small size of the band gap or sub-band energy splitting. On the contrary, we demonstrated that ferroelectricity can be a new mechanism in generating large Berry curvature dipoles and spin-polarized photogalvanic currents in a large gap system without external bias. In this aspect, we are confident that our new scientific result distinguishes itself from previous ones.

We also believe that we have reinforced our argument by reshaping the logical flow and fully reflected all the suggestions and questions raised by the referee into the revised version. We thus hope that the revised manuscript is now suitable for publication in *Nature Communications*.

REVIEWERS' COMMENTS:

Reviewer #2 (Remarks to the Author):

I appreciate authors' careful response and modification of manuscript according to our suggestion. At recent version, I think it reaches the standard of Nat. Comm., and I would recommend it to publish on Nat. Comm. after correcting a mistake:

1). The magnitude and dimension of coefficient in Extended Data Figure 5 seems incorrect.

Reply to Reviewer #2

I appreciate authors' careful response and modification of manuscript according to our suggestion. At recent version, I think it reaches the standard of Nat. Comm., and I would recommend it to publish on Nat. Comm. after correcting a mistake:

1). The magnitude and dimension of coefficient in Extended Data Figure 5 seems incorrect.

(ANS) We would like to thank the reviewer for the recommendation of our paper for publication in *Nature Communications*. We also appreciate that the reviewer carefully brings up an important point to our attention. Following the reviewer's comment, we changed the unit of η in Supplementary Figure 5 (Extended Data Figure 5 in the previous version) by considering the reduced dimensionality of the SnTe monolayer. The charge density (J) is usually defined as current per unit area [A/m^2] in a three dimensional case. However, for a two-dimensional system, J should be changed to current per unit length [A/m] and thus η has a unit of [$nA \ m \ V^{-2} \ S^{-1}$] in the SnTe monolayer, which is independent of the thickness of the supercell.

Once again, we appreciate the reviewer's invaluable comments during the whole review process, indeed leading to the final publication.